# Brain Functional Network and Amino Acid Metabolism Association in Females with Subclinical Depression

**DOI:** 10.3390/ijerph19063321

**Published:** 2022-03-11

**Authors:** Shanguang Zhao, Selina Khoo, Siew-Cheok Ng, Aiping Chi

**Affiliations:** 1Centre for Sport and Exercise Sciences, University Malaya, Kuala Lumpur 50603, Malaysia; s2013074@siswa.um.edu.my; 2Department of Biomedical Engineering, Faculty of Engineering, University Malaya, Kuala Lumpur 50603, Malaysia; siewcng@um.edu.my; 3Institute of Physical Education, Shaanxi Normal University, Xi’an 710119, China

**Keywords:** subclinical depression, phase lag index, complex brain network, brain–gut axis

## Abstract

This study aimed to investigate the association between complex brain functional networks and the metabolites in urine in subclinical depression. Electroencephalography (EEG) signals were recorded from 78 female college students, including 40 with subclinical depression (ScD) and 38 healthy controls (HC). The phase delay index was utilized to construct functional connectivity networks and quantify the topological properties of brain networks using graph theory. Meanwhile, the urine of all participants was collected for non-targeted LC-MS metabolic analysis to screen differential metabolites. The global efficiency was significantly increased in the α-2, β-1, and β-2 bands, while the characteristic path length of β-1 and β-2 and the clustering coefficient of β-2 were decreased in the ScD group. The severity of depression was negatively correlated with the level of cortisone (*p* = 0.016, *r* = −0.40). The metabolic pathways, including phenylalanine metabolism, phenylalanine tyrosine tryptophan biosynthesis, and nitrogen metabolism, were disturbed in the ScD group. The three metabolic pathways were negatively correlated (*p* = 0.014, *r* = −0.493) with the global efficiency of the brain network of the β-2 band, whereas they were positively correlated (*p* = 0.014, *r* = 0.493) with the characteristic path length of the β-2 band. They were mainly associated with low levels of L-phenylalanine, and the highest correlation sparsity was 0.11. The disturbance of phenylalanine metabolism and the phenylalanine, tryptophan, tyrosine biosynthesis pathways cause depressive symptoms and changes in functional brain networks. The decrease in the L-phenylalanine level may be related to the randomization trend of the β-1 frequency brain functional network.

## 1. Introduction

There are about 100 trillion microorganisms with different forms and functions in the human intestinal flora, which is an essential part of the organism [1]. The intestinal flora involved not only in regulating the complex processes of organism physiology, but also affects the function of the central nervous system by mediating the vagus, immune system, and endocrine system [2]. Studies have shown that the intestinal flora of patients with depression is significantly different from that of the normal population and can affect brain function through the gut–brain axis, indicating that the intestinal flora is closely related to the occurrence and development of depression [3].

The brain–gut axis is a regulatory system for bidirectional signal transmission be-tween the brain and the gastrointestinal tract. Studies have shown that depression and gastrointestinal disorders have a higher co-morbidity [4]. Depression may cause intestinal disorders that could result in neuroendocrine and enteric nerve disorders through the brain–gut axis, resulting in an imbalance of related metabolites in the body. Previous studies have shown that imbalance in metabolites such as glutamate–glutamine [5], phenylalanine [6], tyrosine [7], and alanine [8] are related to persons with depression. Some of the metabolites from the gut are absorbed into the circulation and eventually chemically modified (that is, co-metabolized) by the host, and finally excreted with the urine [9]. Metabolomics is a powerful technique that simultaneously detects hundreds of small molecules present in a given biological system, such as fecal, urine, or saliva samples [10]. Brain networks, such as sensorimotor, emotion regulation, dominance, and executive control networks, have been identified to be associated with the pathophysiology of major depressive disorder (MDD) and hypervigilance associated with intestinal disease symptoms [11]. Although many studies have shown that amino acid metabolism is disturbed in persons with depression, the relationship between amino acid metabolism and brain functional network in depression remains unclear.

Complex brain network analysis based on graph theory suggests that the topological disorder of large-scale functional and structural brain networks in depression may be used as biomarkers for the early detection of depression [12]. A recent study has shown that the functional network of the brain has been altered in subjects with subclinical depression (ScD) when compared to healthy subjects [13]. Depression is associated with the abnormal topology of complex brain networks, including global properties and regional connectivity heterogeneity [14]. Li et al. found that the complex brain network of patients with depression showed a trend of randomization in emotion processing, and the abnormal topology of the neural network appeared in the anterior frontal lobe and occipital lobe [15]. Shim et al. found that the brain complex network attributes (including intensity, clustering coefficient, path, and efficiency) of patients with depression changed compared with the healthy group, suggesting that a disorder of the brain complex network index based on EEG may reflect the changes in the emotion processing of patients with MDD [16].

ScD is considered to be prodromal/premorbid to MDD and has become widespread among college students worldwide in recent years [17]. Female college students are more affected by stress and anxiety than males, leading to a higher prevalence of subclinical depression [18]. However, no studies have reported the relationship between gut flora metabolism and functional brain networks in subclinical depression. For female individuals with ScD, do gut microbes influence the production of depressive mood and alterations in functional brain networks through intermediate metabolites? If present, how are the altered levels of metabolites related with the brain function network?

Therefore, in this study, we first investigated the differences in functional brain networks between female students with and without ScD using graph theory. Then, we identify the changes of endogenous metabolites and metabolic pathways in the urine of female students with ScD by using liquid chromatography–mass spectrometry (LC-MS). Finally, the relationship between depressive, functional brain networks and metabolic substances were further studied based on the above findings.

## 2. Materials and Methods

### 2.1. Participants

Forty first-year female college students with ScD and thirty-eight healthy female controls (HC) were recruited into the study. They were aged between 18 and 19 years (mean + SD; 18.51 + 0.42 years). Participants were first-year female students at Shaanxi Normal University. They were assessed by two psychologists from the University Counseling Center using the Self-Rating Depression Scale (SDS) [19] and the Beck Depression Scale-II (BDI-II). The Chinese BDI-II is a 21-item self-reporting inventory used to assess depressive symptoms, and it has good reliability (α = 0.911) [20]. Inclusion criteria for the ScD group were BDI-II scores between 14 and 27 [21]. The exclusion criteria included: 1. A history of traumatic brain injury or depression; 2. Have attempted suicide; 3. Have taken psychiatric medications (including antidepressants, mood stabilizers, antipsychotics, and benzodiazepines); 4. Have physical comorbidities (e.g., cerebrovascular disease and cancer); 5. Have mental comorbidities (e.g., schizophrenia, bipolar disorder, and post-traumatic stress disorder). According to the Edinburgh Handedness Inventory, all participants were right-handed to reduce differences in brain activation. The demographic data of participants are shown in Table 1. There were no statistical differences for age, BMI, and education level between the two groups. The two groups had significantly different mean BDI-II and SDS scores, at *p* < 0.001.

The study was conducted following the Declaration of Helsinki, and research ethical approval was obtained from the Ethics Committee of Shaanxi Normal University. All participants gave written informed consent and received financial compensation for participation in the study.

### 2.2. Metabolomics Analysis

Urine was collected for untargeted LC-MS metabolic analysis. Before collecting urine samples, the participants were given a standard diet for three days and asked to avoid strenuous exercise. Breakfast consisted of eggs, milk, steamed buns, and porridge. Lunch and dinner consisted of two meat and two vegetables + milk or yogurt. Approximately 50% of the energy comes from carbohydrates, 35% fat, and 15% protein in a standard diet. After an overnight fast, the 2 mL morning midstream urine samples were (7:00–8:00 a.m.) collected with a urine collection tube with a lid for all participants and transported in liquid nitrogen tanks to −80 °C cryopreservation.

LC-MS/MS analyses were performed using a UHPLC system (1290, Agilent Technologies, Santa Clara, CA, USA) with a UPLC HSS T3 column (2.1 mm × 100 mm, 1.8 μm) coupled to Q Exactive (Orbitrap MS, Thermo). The mobile phase A was 0.1% formic acid in water for positive, 5 mmol/L ammonium acetate in water for negative, and the mobile phase B was acetonitrile. The elution gradient was set as follows: 0 min, 1% B; 1 min, 1% B; 8 min, 99% B; 10 min, 99% B; 10.1 min, 1% B; 12 min, 1% B. The flow rate was 0.5 mL/min. The injection volume was 1 μL. The QE mass spectrometer was used to acquire MS spectra on an information-dependent basis (IDA) during an LC/MS experiment. In this mode, the acquisition software Xcalibur 4.0.27 (Thermo. Xi’an, Shaanxi, China) continuously evaluated the full scan survey MS data as it collects and triggers MS spectra acquisition depending on preselected criteria. ESI source conditions were set as follows: the sheath gas flow rate was 45 Arb, the aux gas flow rate was 15 Arb, the capillary temperature was 320 °C, the full MS resolution was 70,000, the MS resolution was 17,500, the collision energy was 20/40/60 eV in the NCE model, the spray voltage was 3.8 kV (positive) or −3.1 kV (negative). The processed data were fed into SIMCA + 14.1 software (V14.1, Umetrics AB, Umea, Sweden) for multivariate statistical analysis, including principal component analysis (PCA) and orthogonal projections latent structures–discriminate analysis (OPLS-DA).

### 2.3. Resting-State EEG Recording and Preprocessing

Brain activity was measured from 32 channels following the 10/20 international electrode placement system (Neuroscan Inc., Charlotte, NC, USA). All EEGs were continuously sampled at 1024 Hz and 0.01–100 Hz online bandpass filtering, with the electrodes CPz and AFz used as reference and ground. Vertical electrooculography (EOG) activity was recorded with electrodes placed above and below the left eye; horizontal EOG activity was recorded with electrodes placed on the outboard of both eyes. For all electrodes, the impedance was kept below 10 KΩ during the EEG recording.

The closed-eye EEG data were analyzed to exclude the cortical processing of visual input. The raw EEG data were preprocessed using EEGLAB (Version R2013b, San Diego, CA, USA), and an open-source toolbox running on the MATLAB environment (Version R2013b, MathWorks, Natick, MA, USA). Resting-state datasets were pre-processed with a variety of procedures. A 0.5–45 Hz bandpass filter and a 50 Hz notch filter were applied using a finite impulse response (FIR) filter. Hereafter, the EEG was divided into segments of 2 s duration and reassembled to the averaged reference electrode. The EEGs were down-sampled to 512 Hz. These segments were then visually inspected to remove artifacts (i.e., eye movements, cardiac activity, and scalp muscle contraction) using the independent component analysis (ICA) procedure to identify and extract visual artifact components. Finally, any EEG epochs with amplitude values exceeding ±80 μV at the electrodes were rejected. The Laplacian method was used for the spatial filtering of EEG signals to reduce the influence of the volume conduction effect [22].

### 2.4. Resting-State Brain Network Analysis

#### 2.4.1. Phase Lag Index

The 20 canonical electrodes (FP1, FP2, F7, F3, Fz, F4, F8, T7, C3, Cz, C4, T8, T5, P3, Pz, P4, T6, O1, Oz, O2) of the 10–20 system were selected from the 32 electrodes to construct the brain network. In this study, the PLI was used as a metric for functional connectivity [23]. The 20 channels represent the brain network nodes and the PLI values of the 2-channel signals were used as the weights of the connected edges to construct an unweighted, undirected brain network. A functional connection matrix *C* with seven dimensions of 20 × 20 was obtained.

For any two EEG signals x*_j_* (*t*) and x*_k_* (*t*), if *ϕ*_1_ and *ϕ*_2_ are the phases of two EEG signals, and Δ*ϕ* is the phase difference, the general n to m (with n and m being some integers) phase synchronization can be expressed as:(1)$$\left|\Delta \phi_{n,m}\left(t\right)\right|=\left|n\phi_{j}\left(t\right)-m\phi_{k}\left(t\right)\right|<\;\mathrm{const}\;$$

If the two time series are changing synchronously, then the phase difference will approach a constant. The PLI between the two time series is defined as the value of asymmetry of phase difference distribution:(2)PLI=|〈sign[sin(Δφ(tk))]〉|

#### 2.4.2. Topological Properties of Brain Network

The Brain Connectivity Toolbox was used for graph theory analysis [24]. The magnitudes of the elements C*_i_*_,*j*_ (0 < C*_i_*_,*j*_ < 1) in the matrices represent the PLI of the functional connection between node *i* and node *j*. The quantitative analysis of brain functional networks using graph theory requires thresholding each matrix *C* to create a binary matrix *A*. That is, if the absolute value of the PLI C*_i_*_,*j*_ between node *i* and node *j* is larger than a given threshold T (0 < T < 1), the value of the element of the binary matrix corresponding to the position of *r_i_*_,*j*_ will be set to 1; otherwise, it is set to 0. Each binary matrix defines an unweighted graph *G*. Any node in the graph is connected to other nodes through *k* undirected edges corresponding to non-zero elements (*a_i_*_,*j*_ ≠ 0). This study used a series of consecutive sparsity thresholds *S* to transform the PLI matrices into an array of corresponding binary matrices. On the basis of previous studies, we chose the set of sparsity thresholds with a step size of 0.01 and minimum and maximum values of 0.05 and 0.40, respectively (0.05 < S < 0.40) [25]. As a result, for each subject, a total of 36 brain functional network sets with progressively increased sparsity levels were obtained.

The characteristic path length (L*_p_*), clustering coefficient (C*_c_*), global efficiency (E*_global_*), and local efficiency (E*_local_*) are used in graph theory analysis to quantify the distribution of networks.


(3)
$$Lp=\frac{1}{N\left(N-1\right)}\sum_{i,j\subseteq V,i\neq j}\;d_{ij}$$



(4)
$$Cc=\frac{1}{n}\sum_{i\in N}\;C_{i}=\frac{1}{n}\sum_{i\in N}\;\frac{2t_{i}}{k_{i}\left(k_{i}-1\right)}$$



(5)
$$E_{global}=\frac{1}{n}\sum_{i\in N}\;E_{i}=\frac{1}{n}\sum_{i\in N}\frac{\sum_{j\in N,j\neq i}\;d_{ij}^{-1}}{n-1}$$



(6)
$$E_{local}=\frac{1}{n}\sum_{i\in N}E_{\mathrm{loc},i}=\frac{1}{n}\sum_{i\in N}\frac{\sum_{j,h\in N,j\neq i}\;a_{ij}a_{ih}{\left[d_{jh}\left(N_{i}\right)\right]}^{-1}}{k_{i}\left(k_{i}-1\right)}$$


### 2.5. Statistical Analysis

The Shapiro–Wilk test was used to determine the normality of the data distribution. Continuous variables are expressed as means with SD. Demographics of the HC and ScD groups were compared using Wilcoxon non-parameters. The Network-Based Statistics (NBS) toolbox analyzed differences in brain functional network connectivity between two groups. An independent samples *t*-test with false discovery rate (FDR) correction was used to quantify the topological properties of the brain network. The Cohen’s d was computed to estimate the effect size of independent sample *t*-tests. Spearman’s correlation analysis was used to quantify the potential link of metabolites and topological properties of brain network. The level of significance was set as a 2-sided *p* value less than 0.05.

All compounds were screened for potential differential metabolites using the Kyoto Encyclopedia of Genes and Genomes (KEGG). The variable importance in the projection (VIP) value of each variable in the PLS-DA model was calculated to indicate its contribution to the classification. Metabolites with the VIP > 1 were further applied to Student’s *t*-test at the univariate level to measure the significance of each metabolite, with results adjusted for multiple testing using the Benjamini–Hochberg procedure with the critical FDR set to 0.05. Metabolic pathway analysis and enrichment analysis on these differentiated metabolites was conducted using MetaboAnalyst 5.0. In this study, “Homo sapiens (KEGG)” was selected as the background library. The statistical analyses were performed using SPSS (23.0; SPSS, Inc., Chicago, IL, USA). The GraphPad 8.0 software (Prism. San Diego, CA, USA) was used for visualizations.

## 3. Results

### 3.1. Differential Metabolite Identification

By analyzing the VIP of the first principal component of the OPLS-DA model (VIP > 1) and the *p*-value of Student’s *t*-test (*p* < 0.05), 23 differential metabolites met the screening conditions for the discrimination between the HC and ScD group, of which 7 were screened under the positive ion model and 16 under the negative ion model. The relative change in the above substances is shown in Figure 1A. The pathway impact distribution maps (Figure 1B) show that the three perturbed pathways meet the screening criteria in the ScD for phenylalanine metabolism, phenylalanine, tyrosine, and tryptophan biosynthesis, and nitrogen metabolism.

### 3.2. Resting-State Brain Network Analysis

The topological graphs of the brain functional network were plotted based on the adjacency matrix to reflect the differences in brain functional connections between two groups, as shown in Figure 2. The results show that there are connection differences between HC and ScD groups in delta, alpha2, beta-1, and beta-2 bands (*p* < 0.05, NBS corrected). We, therefore, quantify the topological properties of the brain network in four frequency bands at multiple sparsity levels. The results are shown in Figure 3, where the broken line graph represents the network topology properties of each sparsity. We averaged the thresholds with significant differences. Further analysis showed that the E*_global_* was significantly higher for the ScD group than for the HC group in the α-2 (*t* = 2.25, *df* = 61, *p* = 0.028, Cohen’s d = 0.58), β-1 (*t* = 3.07, *df* = 61, *p* = 0.003, Cohen’s d = 0.79), and β-2 (*t* = 3.02, *df* = 61, *p* = 0.004, Cohen’s d = 0.77) bands (0.53 ± 0.01 vs. 0.50 ± 0.01; 0.48 ± 0.003 vs. 0.46 ± 0.005; 0.63 ± 0.003 vs. 0.60 ± 0.006) (Figure 4). The L*_p_* was significantly lower for the ScD group than HC in the β-1 (*t* = −2.84, *df* = 61, *p* = 0.006, Cohen’s d = 0.73) and β-2 (*t* = −2.81, *df* = 61, *p* = 0.007, Cohen’s d = 0.72) bands (2.17 ± 0.03 vs. 2.3 ± 0.004; 1.67 ± 0.001 vs. 1.75 ± 0.003) (Figure 4). The C*_c_* of the beta-2 band in the ScD group (*M* = 0.48, *SE* = 0.01) was smaller than that in the HC group (*M* = 0.52, *SE* = 0.01) with a network sparsity of 0.34–0.36 (*t* = −2.26, *df* = 61, *p* = 0.027, Cohen ‘s d = 0.59) (Figure 4). The result is shown in the bar chart in Figure 4.

### 3.3. Correlation of Metabolite and EEG Network Properties

By analyzing the correlation between the changing urinary metabolite levels and the BDI-Ⅱ score, the results showed a negative correlation between the severity of depression and cortisone levels (*p* = 0.016, *r* = −0.40). We also analyzed the correlation between the topological properties of the brain network and differential metabolites. The biosynthesis of tyrosine, tryptophan, phenylalanine, and phenylalanine metabolism were negatively correlated (*p* = 0.014, *r* = −0.493) with the E*_global_* of the β-1 band, whereas they were positively correlated (*p* = 0.014, *r* = 0.493) with the L*_p_* of the β-1 band. It was mainly associated with increased L-phenylalanine, and the highest correlation sparsity thresholds were 0.11. The result is shown in Figure 4.

## 4. Discussion

### 4.1. Altered Functional Brain Networks

The basic idea of complex brain network analysis was to identify differences in the topological properties of brain networks between female students with and without ScD using graph theory. The results showed that E*_global_* of functional brain network in the ScD group was significantly higher in the α-2, β-1, and β-2 bands, while the L*_p_* was lower in the β-1 and β-2 bands, and C*_c_* was decreased in the β-2 band. L*_p_* represents the connectivity effect of the global network, which is the effect related to the average of the shortest path connecting all nodes in the network to other nodes. The larger the L*_p_* value, the higher the global transmission efficiency. C*_p_* represents the clustering situation of the local network. The larger the C*_p_* value, the faster the local information is transmitted and the stronger the ability to process information. In conclusion, these results suggest that the shortest path lengths between nodes in the functional brain network of female students with ScD become shorter, that is, the number of edges passed by one node to reach another node becomes smaller. The information carried by the nodes is transmitted more efficiently at the global level, suggesting an increased functional integration within their brains.

The ScD group had a smaller C*_c_* than the HC group, indicating a reduced degree of node aggregation in their functional brain networks, that is, a decrease in the density of connections between nodes. A recent EEG-based study revealed that the subclinical depression group had a lower C*_c_* and larger L*_p_* than healthy controls [13]. The results of C*_c_* are consistent with this study, but the results of L*_p_* are the opposite. It may be that the study required participants to complete an emotion recognition task, whereas we collected EEG data in the rest state, so the increased L*_p_* of the functional brain network may be used for information transmission.

The same results were confirmed in Leistedt et al.’s study, which found a significant increase in the L*_p_* of the brain networks of acutely depressed patients by verifying the relationship between depression and the information processing capacity of neural networks [26]. Several studies also indicate functional brain network disruption in patients with MDD via functional magnetic resonance imaging (fMRI). Zhang et al. found that the functional brain network of patients with first-episode and untreated MDD had a lower L*_p_* and higher E*_global_*, but there was no significant difference in C*_c_* [27]. H. Li et al. also discovered that the L*_p_* of the brain functional network of patients with MDD decreased and the E*_global_* increased, while the C*_c_* was significantly lower [28]. The above results demonstrate that altered functional brain networks in subclinical and clinical depression lead to inefficient information dissemination.

One of the core symptoms that MDD manifests is a loss of interest and expressions of indifference to surrounding emotions [29]. The β-band rhythm mainly occurs in the frontal lobe, related to cortical excitability, and it reflects emotional and cognitive processes [30]. The frontoparietal network, the executive control network, is mainly involved in the advanced cognitive regulation of negative emotional conflicts [31,32]. The topography of brain network connectivity showed that the functional connectivity in the β-band was weakened in the frontal and parietal lobes of female college students with ScD. Kaiser et al. showed that functional connectivity within the frontoparietal control network was significantly reduced in patients with DMM [33]. The reduced L*_p_* and increased E*_global_* of functional brain networks in the β-band of the ScD group imply the functional brain networks associated with depressive mood regulation have been disrupted, with a tendency to shift to random networks. Random networks have low modular information processing or fault tolerance [27]. The randomization process has been observed in the functional brain networks of patients with other neurological or psychiatric disorders, such as Alzheimer’s and schizophrenia [34].

### 4.2. Disorder of Amino Acid Metabolism

This study found that phenylalanine metabolism and phenylalanine, tryptophan, and tyrosine biosynthetic pathways were disrupted in the ScD group, which was mainly associated with a decrease in L-phenylalanine. Studies of urine metabolomics in patients with MDD have also found a trend towards lowering phenylalanine [8,35]. The results of this study are consistent with these findings, suggesting that phenylalanine metabolism is abnormal in the subclinical stage of MDD.

Phenylalanine, tyrosine, and tryptophan are all aromatic amino acids [36]. Phenylalanine is a precursor for synthesizing tyrosine and catecholamines, which is primarily catabolized in the liver and catalyzed by phenylalanine hydroxylase [5,37]. The reduction of L-phenylalanine levels in female students with ScD may be due to decreased phenylalanine hydroxylase enzyme activity [6]. The catecholamine neurotransmitters such as 5-hydroxytryptamine (5-TH), dopamine, adrenaline, and norepinephrine are synthesized from phenylalanine and tyrosine through hydroxylation and decarboxylation reactions [38]. Numerous studies have found that the concentration of monoamines (such as 5-TH, norepinephrine and dopamine) in the synaptic gap drops in depressive states [39]. The most critical metabolism of tyrosine is the production of dopamine catalyzed by tyrosine hydroxylase. Dopamine is produced in the presence of dopa decarboxylase. Dopamine is the richest catecholamine neurotransmitter in the brain and regulates many physiological functions of the central nervous system. Dopamine can also be produced via the action of dopamine β-hydroxylase to produce norepinephrine, which in turn is produces adrenaline through the action of methyltransferase. These results revealed that a fall in L-phenylalanine levels in female college students with ScD might cause disruptions of tyrosine synthesis and monoamine neurotransmitter levels, such as 5-TH and dopamine, might tend to decline.

Tryptophan is metabolized mainly in the brain, liver and intestine, among which the colon is the main site of tryptophan intake in the body. The starting amino donor for tryptophan biosynthesis is glutamate or ammonia [40]. In this study, a significant reduction in ammonia levels was found in students with ScD. Tryptophan, the unique raw material for 5-TH synthesis, has also been highly relevant to the development of depression [41]. More than 90% of human 5-HT is produced in the intestine, and bacteria in the intestine influence 5-HT production by expressing tryptophan synthase [42]. Since 5-HT does not cross the blood–brain barrier, the central and peripheral are two separate systems. Here, 5-HT, an important neurotransmitter in the central nervous system, may lead to abnormal 5-HT receptor function and neurotransmission disorders if abnormalities occur in the central nervous system, and changes in its level are often associated with mood disorders and depression [43]. Studies have shown that the dysregulation of tryptophan metabolism is a possible mechanism for depression-related behaviors [44]. Booij and Van der Does found that acute depletion of tryptophan results in depressive symptoms, which may be caused by a dramatic decrease in 5-HT production [45]. Maes et al. proposed that the decrease in 5-HT levels in depressed patients probably comes from the activation of the tryptophan-metabolizing enzymes TDO and IDO, leading to depletion of tryptophan in the plasma and consequently to a drop in 5-HT levels in the brain [46]. Peripheral 5-HT is involved in vasoconstriction and vasodilation, metabolic rate changes, temperature control, inflammation, and fibrosis [47]. It is shown that altered levels of 5-HT may also cause the altered levels of inflammatory cytokines involved in depression due to changes in the microflora [48].

In conclusion, the findings demonstrate that disorders of tryptophan biosynthesis in the peripheral system of female college students with ScD may be due to insufficient tryptophan synthesis caused by a decrease in ammonia levels. However, insufficient tryptophan synthesis triggers a decrease in 5-TH levels in female college students with ScD, which in turn leads to depressive disorders.

### 4.3. The Relationship between Amino Acid Metabolism and the Functional Brain Network

The metabolism is a critical pathway through which intestinal flora affects depression via the brain–gut axis by directly altering the levels of key metabolites or indirectly altering circulating serum metabolites, which can modulate depressive behavior in the central nervous system [49]. Disturbances in tryptophan metabolism may be partly responsible for the mood, cognitive, and sleep disturbances typical of depression. The 5-hydroxytryptamine theory of depression states that decreased 5-hydroxytryptamine release in the central nervous system and its reduced levels in the synaptic gap are key contributors to depression [2]. Animal studies found that the colonization of the intestinal flora of depressed patients in the germ-free rat intestine caused altered neurobehavior and increased the ratio of plasma kynurenine to tryptophan [50]. Alterations in tryptophan metabolism may be driven by intestinal flora, leading to depressive symptoms. The flow chart of the relationship between the functional brain network and the peripheral metabolic system is shown in Figure 5.

This study showed that phenylalanine levels were negatively correlated with E*_global_*, whereas they were positively correlated with the L*_p_* of the β-1 band of the functional brain network in female college students with ScD. Given the above discussion, we found that the β-1 band of brain functional network associated with negative emotion regulation in female college students with ScD tends to be randomized, while the disruption of phenylalanine metabolism and phenylalanine, tryptophan, and tyrosine biosynthetic pathways is mainly associated with a decrease in L-phenylalanine. In conclusion, the disturbance of phenylalanine metabolism and the phenylalanine, tryptophan, tyrosine biosynthesis pathways cause depressive symptoms and changes in functional brain networks. The decrease in the L-phenylalanine level may be related to the randomization trend of the β-1 frequency brain functional network.

## 5. Conclusions

On the basis of the complex brain network, this study investigated the differences in the topological properties of the brain network between female college students with and without ScD. The results showed that the β-1 band functional network tended to be random, which might reflect emotional processing changes. The decrease in the L-phenylalanine level may be related to the randomization trend of the β-1 frequency brain functional network. The disturbance of phenylalanine metabolism and phenylalanine, tryptophan, and tyrosine biosynthesis pathways may lead to depressive symptoms and changes in functional brain networks.

## Figures and Tables

**Figure 1 ijerph-19-03321-f001:**
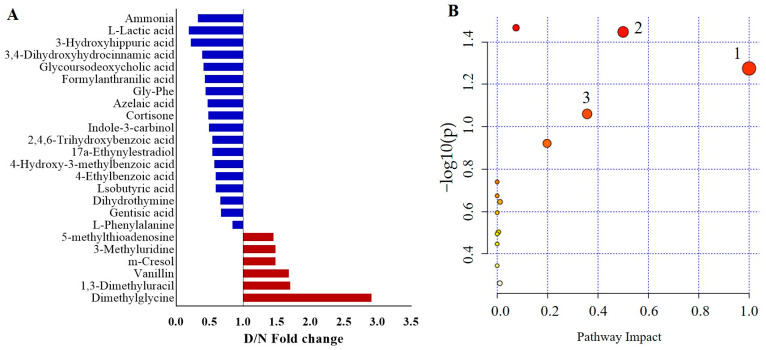
Differential metabolites (**A**) and metabolic pathways (**B**) between ScD and HC groups; Note: FC > 1 means the metabolite is up-regulated in ScD. FC < 1 means the metabolite is down-regulated; 1, nitrogen metabolism; 2, phenylalanine, tyrosine, and tryptophan biosynthesis; 3, phenylalanine metabolism.

**Figure 2 ijerph-19-03321-f002:**
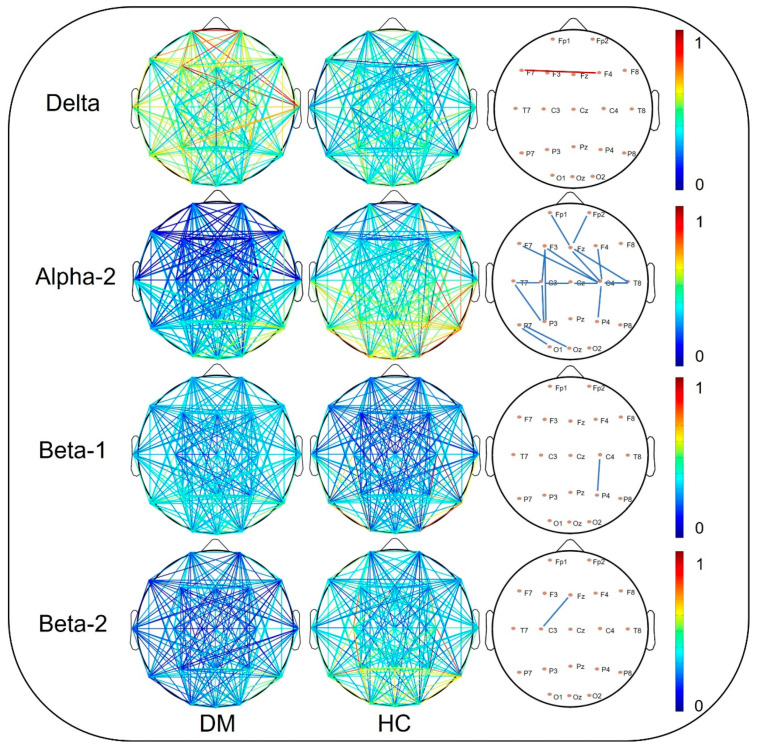
Brain network connectivity graphs of ScD and HC groups; on the right is the difference network diagram (red: ScD > HC; blue: ScD < HC); the strength of the connection between the nodes is indicated by the color of the line.

**Figure 3 ijerph-19-03321-f003:**
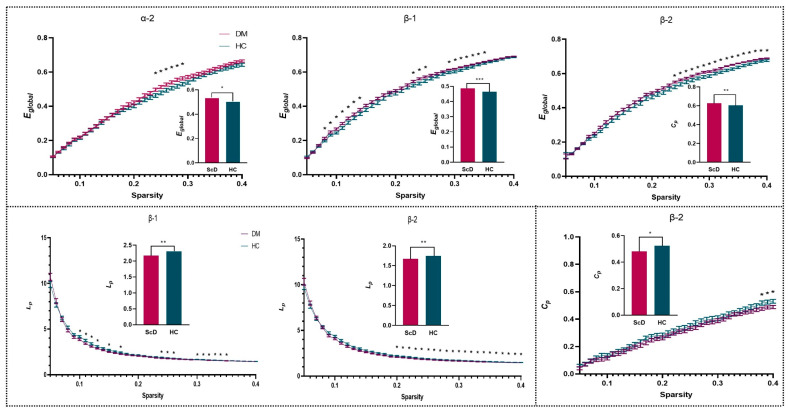
Topology properties of brain network based on sparsity threshold (0.05 < S < 0.4); the bar chart represents a comparison after averaging all thresholds with significant differences. * *p* < 0.05. ** *p* < 0.01.

**Figure 4 ijerph-19-03321-f004:**
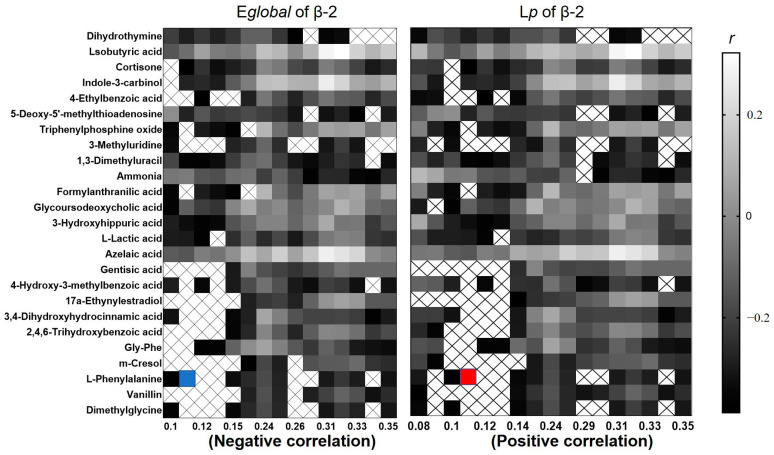
Correlation between E*_global_* and L*_p_* of functional brain network and differential metabolites; × represents a significant correlation. The maximum correlation (red and blue square) was found at the sparsity of 0.11.

**Figure 5 ijerph-19-03321-f005:**
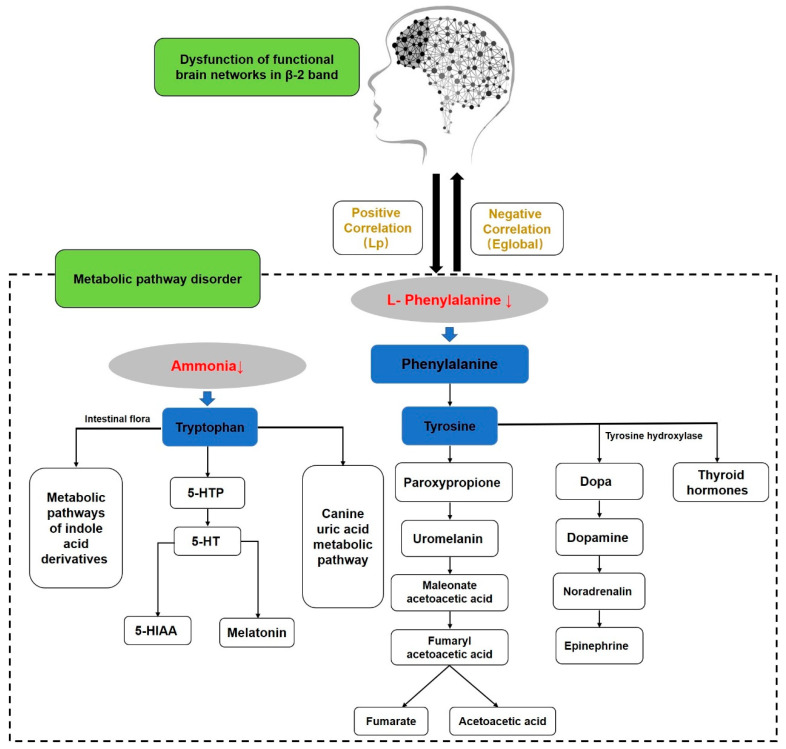
The relationship between brain function network and peripheral metabolic system.

**Table 1 ijerph-19-03321-t001:** Demographic information of participants (Mean ± SD).

Variable	ScD (*n* = 40)	HC (*n* = 36)
Age, years	18.72 ± 0.36	18.51 ± 0.42
Height, cm	162.71 ± 6.62	160.70 ± 6.73
Weight, kg	52.37 ± 4.72	50.00 ± 1.92
BMI, kg/m^2^	20.79 ± 2.73	19.43 ± 1.61
SDS	10.57 ± 5.47	66.71 ± 5.38 ***
BDI-II	3.46 ± 0.73	24.86 ± 2.02 ***

Note: HC, healthy controls; ScD, subclinical depression; BMI, body mass index; SDS, Self-rating Depression Scale; BDI-Ⅱ, Beck Depression Inventory Ⅱ; *** *p* < 0.001.

## Data Availability

The study data can be accessed from the author S.Z. by request.

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
