# Peer review of "Brain Functional Network and Amino Acid Metabolism Association in Females with Subclinical Depression"

_ijerph, 2022, doi:10.3390/ijerph19063321_

Round 1

Reviewer 1 Report

Line 12  - Spell out EEG.

Line 51 – Spell out MDD.

Line 57 – The word “development” is not clear. Do the authors mean fetus development of the central nervous system or are referring to adults? If so, what kind of development?

Line 61 – Despite included in the discussion, I suggest mentioning very briefly here that 5-hydroxytryptamine is serotonin, and that is derived from the essential amino acid. Tryptophan.

Line 130 – Please provide a small explanation of  the relevance of considering that “all participants were right-handed”.

Line 136 – It is mentioned that the participants “received financial compensation”. Assuming that some of them were bellow 18 years old, I wonder if parents were included in this process. This should be referenced.

Line 143 – I understand that the purpose of giving a standard diet for three days was to reduce diet induced bias intra- and intergroups. However, which diet? What criteria was used to choose 3 days?

Line 341 – I wonder if it is correct considering MDD in the discussion so extensively. The focus of this study is ScD with only data form ScD. MDD is certainly not only a “more intense” form of ScD in terms of pathways and metabolomics, and thus exactly comparable with ScD. This limitation should be referred and additionally, including other literature on ScD would be certainly better.

Reviewer 2 Report

The proposal to analyze the links of  amino acid metabolism to brain EEG (so called  "Brain functional network" , but the study design did not reflect to the gut-brain axis.
in addition, this study only included female but not in comparison to male. therefore, it is hard to confirm whether the observations only observed in female.

In section of Abstract 
Line18:  "Results "The severity of depression was negatively correlated with the level of cortisone (p = 0.016, r = -0.40)." however, it is not clear what's the purpose of this assay. In fact, as early as 2009, cortison is routinely used as a biomarker of psychological stress and depressive symptoms by Hellhammer et al. Psychoneuroendocrinology 2009 34 163171. (doi:10.1016/j.psyneuen.2008.10.026)

in the beginning of abstract and also in section of introduction, "intestinal flora" was emphasized, but in this study,  urine metabolites were reported.
Lines 24-27: the following results were described " It was mainly associated with increased L-Phenylalanine, and the highest correlation sparsity was 0.11." BUT, authors also concluded that " The decrease of L-phenylalanine level may be related to the randomization trend of the β-2 frequency brain functional network". it is confusing.

in abstract, methods regarding how to determine and how those participants were recruited as subclinical depression need to be described.

Descriptions at lines 60-61 also suggested that dietary behaviors whether intaking  Bifido... may changes plasma levels of amino acids.

The paragraphs mentioned about intestinal flora is not directly to urine amino acids analysis can be deleted. It is recommended that focus o the urine amino acids linked to EEG data and also recruit male students as volunteers.

Round 2

Reviewer 2 Report

Thanks for the revision and responses from authors to find comments valued.

however,  regarding the rationale descriptions still not clear. for example, lines 9-11, 31~77 described so many references about flora, but in current study did not analyze the gut microbiota. therefore, the references which  describing urine amino acids linked to the gut flora would be very important the point out the background information.  such as the following reference : https://www.nature.com/articles/s41598-019-45640-y.pdf?proof=t%2Btarget%3D

It seems to be much likely to be cited as the background to introduce the identified amino acids linked to floea which authors can discuss those known to be involving brain functions.

so readers can be directed into the purpose of this study, and authors also can find some supporting from those articles investigating about metabolites either from blood or urine and then explaining their link to brain networks. 

Round 3

Reviewer 2 Report

Authors improved the writings greatly including abstract and also the parts in red, but the current introduction section can be more concisely. please keep most relevant references which was inserted paragraphs and try to to delete some of the paragraphs. The same for the analytical methodology. If it was not developed by current study, please try to cite the references of utilized methods, and delete the details of methodology descriptions in order to have full contents to be focused on the manuscript findings which would be appreciated. Format of listed references need to be modified according to the journal's instructions. 
